# Position: Natural Language Should Not Fully Replace Formal Languages

Eitan Wagner [1] [2]  Elisha Rosensweig [2]  Omri Abend [1]

## Abstract

Recent advances in large language models and their widespread adoption have prompted claims that natural language could entirely replace formal languages, such as programming languages for software design. In this position paper, we argue that this perspective overlooks fundamental linguistic properties of natural language, specifically that it is optimized for underspecification in open-ended contexts. We introduce a formal framework centered on *task specificity*, defining it as the information-theoretic reduction of uncertainty in an output space—such as all possible images—given a user's specific requirements. We prove a *specificity crossover theorem*, showing the existence of a threshold beyond which the cost to express formal requirements into natural language exceeds the cost of direct formal specification. By analyzing case studies across modalities, such as image generation, code synthesis, and audio production, we demonstrate that natural language excels at low specificity tasks, while formal languages are advantageous on tasks with stricter requirements. We conclude that natural and formal languages are complementary tools and advocate the development of hybrid systems that allow users to move across the specificity spectrum.

## 1. Introduction

Large language models (LLMs) are transforming human-technology interaction, raising questions about the future of specialized expertise. As these models demonstrate increasing capabilities, concerns about professional skill displacement intensify. Programming represents a particularly salient case study: code is economically significant, operates in well-defined formal domains, and benefits from abundant training data (Jiang et al., 2024; Joel et al., 2024). AI for coding is massively adopted—84% reported using AI tools in the Stack Overflow 2025 survey,[1] with increasingly positive attitudes toward adoption (Eshraghian et al., 2025). Some envision a future where natural language entirely replaces programming languages—a vision of "The new programming language is called Human" (Jensen Huang; London Tech, June 2025).[2] However, current systems face substantial limitations: struggles with domain-specific code (Joel et al., 2024), difficulties adapting to changing requirements (Jiang et al., 2024), and poor security (Dora et al., 2025). In general, more developers report active distrust (46%) than trust (33%) in the accuracy of AI tools.[3] These challenges raise questions about whether they represent temporary engineering hurdles or fundamental limitations.

**Our Position.** We argue that natural language and formal languages (such as the code itself) serve complementary information-theoretic roles, optimized for different points along what we call the *ladder of specification*—a hierarchy we define building on previous research on communication efficiency and ambiguity in natural language (Gibson et al., 2019; Piantadosi et al., 2012). Natural language addresses *open-endedness*—the challenge of communicating about an unbounded world—through underspecification (Roberts, 2012), which includes *pragmatic underspecification* (where listeners infer the speaker's intent from context) and *indifferent underspecification* (where the speaker genuinely accepts multiple outcomes). Formal languages sacrifice this flexibility for precise description within restricted domains, operating at *full specification* where every relevant detail is explicit. The cost of crossing between these modalities—the *translation gap*—means that neither approach dominates: natural language wins at low task specificity through underspecification, while formal languages win at high task specificity by reducing the risk of errors and avoiding a translation overhead. This complementarity suggests that research should focus not only on formal language generation capabilities but also on effective human-AI collaboration, recognizing when each modality is advantageous.

[1]Department of Computer Science, Hebrew University of Jerusalem [2]DICTA. Correspondence to: Eitan Wagner <eitan.wagner@mail.huji.ac.il>.

*Proceedings of the 43rd International Conference on Machine Learning*, Seoul, South Korea. PMLR 306, 2026. Copyright 2026 by the author(s).

[1]https://survey.stackoverflow.co/2025/ai#sentiment-and-usage
[2]https://youtu.be/SsNS3Xm9ig4?t=1940
[3]https://survey.stackoverflow.co/2025/ai#developer-tools

## 1.1. Key Contributions

Building upon linguistic insights (§2), we provide an information-theoretic framework (§3) for modeling tasks as constraint satisfaction with explicit and implicit constraints, while distinguishing types of underspecification along a continuum. We prove that for sufficiently specific tasks—and a formal language without exceedingly high redundancy—natural language descriptions become less efficient than direct specification in the target modality. Our analysis validates trends reported in earlier work on many modalities, including images, code, audio, and text (§4).

As a consequence of our analysis (§5), we argue that development and evaluation of code generation from natural language requires a hybrid approach. Consequently, the practical value of end-to-end coding models and agents should also be evaluated in terms of the actual benefit for a workflow, with various hybrid models as baselines, and in both low and high expertise domains.

## 2. Communication Efficiency: Open-Endedness and Underspecification

In this section, we provide linguistic and information-theoretic background regarding underspecification. For the purposes of this paper, we focus on one type of speech act (Searle, 1969): describing—directly or indirectly—a goal to be implemented.

### 2.1. Focusing Communication

Natural language serves as humanity's primary communication tool, optimized for efficient information transfer across diverse contexts (Zipf, 1949). A fundamental challenge is *open-endedness*: natural language must handle the practically unbounded complexity of the real world through the ability to form novel utterances (Hockett & Hockett, 1960; Deacon, 1998). Because language users themselves contribute to this complexity—creating new concepts, cultural practices, and communicative needs—the space of possible meanings is essentially unbounded (Tomasello, 2009).

The Question Under Discussion framework (QUD; Ginzburg et al., 1996; Roberts, 2012) addresses this challenge by viewing discourse as organized around implicit questions that focus communication. The QUD determines what aspects of the world are relevant, licensing the omission of irrelevant details. This mechanism enables natural language to address open-endedness: each speech act specifies only what matters for the current communicative goal.

### 2.2. The Ladder of Specification

Within a given domain and QUD, utterances vary in how precisely they constrain the space of acceptable meanings.

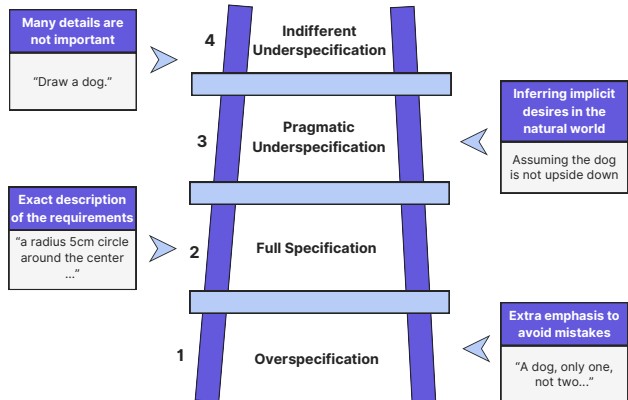

*Figure 1.* The Ladder of Specification: from genuine indifference (top) to overspecification (bottom), with corresponding compression benefits and error risks.

Relaxing constraints over possible meaning is termed *semantic underspecification*. In semantics, multiple frameworks attempt to represent the "shared" meaning of a sentence through a single, compact meta-structure (Reyle, 1993; Copestake et al., 2005). From a cognitive perspective, this approach aligns with the observation that human listeners frequently maintain an underspecified mental representation of ambiguous input, only committing to a specific interpretation when prompted by pragmatic cues or discourse requirements (Poesio, 1995; Frisson, 2009).

We characterize this variation through the *ladder of specification* (see Fig. 1), a hierarchy reflecting different relationships between what is said and what is meant:

**Definition 2.1** (The Ladder of Specification). Communication achieves compression through a specification hierarchy:

1. **Overspecification**: More information is provided than strictly necessary, including redundancy or emphasis (e.g., "a cat, not a dog"). While seemingly inefficient, redundancy can reduce error rates in noisy channels (Shannon, 1948). Even seemingly precise utterances like "at 3 o'clock" allow some implicit flexibility; adding "exactly" reduces the space of possible interpretations, disallowing approximations (Lasersohn, 1999).

2. **Full Specification**: Every relevant detail for the current domain is explicitly stated (e.g., code or exact RGB values for colors). At this level, the description of the intent effectively coincides with its implementation. Formal languages typically operate at this level within their domains (Meyer, 1993).

3. **Pragmatic Underspecification**: The meaning is expected to be inferred from context, where the speaker has intentions beyond what is literally expressed. This level encompasses several distinct situations:

- *Clear intention*: The speaker has a specific outcome in mind but relies on shared knowledge for disambiguation (e.g., "professional styling" assumes standard design principles).
- *Latent preferences*: The speaker initially has no conscious preference but discovers, through interaction and feedback, that they do care about certain properties. The speaker's constraints often emerge through iterative refinements of their expressed goals in a trial-and-error cycle.
- *Delegation*: The speaker does not explicitly formulate preferences for certain details, trusting the listener to make good choices on their behalf (e.g., a manager saying "handle the deployment" trusts the engineer's judgment). Importantly, not all choices are acceptable; the speaker adopts the listener's judgment.

4. **Indifferent Underspecification**: True indifference to output properties, where multiple interpretations are equally acceptable to the speaker (e.g., "use any reasonable color scheme").

The key distinction between pragmatic underspecification and indifferent underspecification lies in whether the speaker has (possibly implicit) preferences about the underspecified properties. In pragmatic underspecification, miscommunication represents a failure; in indifferent underspecification, variation is acceptable by design.

Formal languages, such as programming languages, achieve precision by operating primarily at the full specification level within their restricted domains.[4] Importantly, formal languages sometimes exhibit hierarchical abstraction: the command $x = 1+2$ specifies precisely what value is stored but not where in physical memory or how the processor executes the addition. The formal language is deterministic and unambiguous at the outcome level, while delegating implementation details to lower system levels that have a negligible effect on the output.

### 2.3. Linguistic Perspectives

The ladder of specification connects to several research traditions in linguistics and cognitive science.

**Pragmatics and Inference.** The study of how context shapes meaning beyond literal semantics is central to understanding pragmatic underspecification. Grice (1975) established foundational principles of cooperative communi-

cation, where speakers implicate meanings beyond what they literally say. Levinson (2000) provides a comprehensive treatment of how listeners infer pragmatic meaning. Carston (2008) argues that the actual meaning expressed is always much richer than the words used, with listeners filling in substantial implicit content. Clark (1996) emphasizes the role of "common ground" in this inference process.

The Rational Speech Acts (RSA) framework (Frank & Goodman, 2012; Goodman & Frank, 2016) models pragmatic inference as bounded rational reasoning (Degen, 2023), where speakers and listeners reason about likely interpretations given computational constraints. Attempts to formalize listener inference include *default reasoning* in logic (Reiter, 1980; Mercer, 1987), although others argue that natural situations are too complex for formal logic (Cadoli & Schaerf, 1993) and probabilistic methods better describe the cognitive process (Oaksford & Chater, 2007).

Pragmatic underspecification introduces miscommunication risk. The probability of correctly inferring implicit constraints depends on both the nature of the constraints and the quality of shared context (Ferreira, 2008). Crucially, for successful pragmatic inference, the listener must understand the specific speaker's context, knowledge, and preferences (Bergen & Grodner, 2012). This quality of *personalization* forms an active research topic in dialogue systems (Zadrozny et al., 2000; Oruche et al., 2025).

**Communication Efficiency.** A large body of research addresses communication efficiency through the lens of information theory (Futrell & Hahn, 2022), often through rate-distortion theory, which provides a basis for lossy compression (Shannon et al., 1959). A general finding is that language optimally balances simplicity (shorter messages) and informativeness (Kemp & Regier, 2012; Zaslavsky et al., 2017; Marzen & DeDeo, 2017; Gibson et al., 2019). Krifka (2007) shows that speakers use imprecise language (such as approximate numbers) as a means of reducing effort when precision is unnecessary. Piantadosi et al. (2012) demonstrate that ambiguity serves as a compression device when context enables disambiguation.

Underspecification in communication is fundamentally related to the notion of *autonomy* given to the listener. We expand on this in Appendix A.

**Translation and Specification Mismatch.** Languages differ in what they require speakers to specify: for example, a language with grammatical gender marking forces gender specification (Savoldi et al., 2021; 2025). To preserve underspecification during translation, clarifications may be necessary, potentially making the translation longer than the original. For instance, describing a nurse's actions without specifying gender is natural in English but requires explicit

---

[4]Generally speaking, a formal language is defined by a set of deterministic rules, both for its syntax (i.e., the utterances) and its semantics (i.e., the mapping between utterances and objects). In particular, there is no ambiguity or randomness; an utterance may denote a set of objects, but the mapping itself is not random.

disambiguation in languages with grammatical gender—or an explicit statement that gender is unspecified.

This phenomenon generalizes beyond natural languages: translating from natural language to formal specification often requires resolving underspecification, either by making constraints explicit or by choosing among alternatives the speaker was indifferent to. An example is mathematical reasoning, where formal notation forces rigor and minimal ambiguity. Fully describing mathematical proofs in natural language will lead to long, and often inconsistent, proofs.

**Compositionality.** *Compositionality*—the principle that the meaning of a complex expression is determined by the meanings of its parts and how they combine (Partee et al., 1995; Pelletier, 1994)—enables efficient specification by allowing constraints to be stated independently. "A red sports car" comprises color and type constraints separately, and each component can be identified as the QUD.

The compositional structure of both natural and formal languages enables their combination: specifications can mix languages, using natural language for high-level intent and formal notation for precise details, with compositional boundaries determining where each is most efficient.

# 3. Formal Framework

We formalize these intuitions using a constraint satisfaction framework, inspired by Fregean semantics (Frege et al., 1951), distinguishing between explicit and implicit constraints. We then formalize when natural language loses efficiency to formal specification. Our analysis follows the assumption that shorter utterances are more efficient (Goodman & Frank, 2016; Gibson et al., 2019).

## 3.1. Tasks as Constraint Satisfaction

We address tasks where the input is a textual description, and the output can be from various modalities (images, code, music, etc.). We refer to the space of possible outputs as a "world model"—a confined subset of the real world with well-defined structure—which is specified in a formal language. Adapting notation from formal logic (Partee et al., 2012) and the Rational Speech Act (RSA; Goodman & Frank, 2016) terminology, $U$ denotes the utterance space and $M$ the meaning space.

**Definition 3.1** (Task and Constraints). A task $T$, given as textual description $u \in U$, is represented by a tuple $(C_{\exp}, C_{\imp}, M)$ where:

- $M$ is the space of possible outputs (e.g., all valid images, programs, etc.).

- $C_{\exp}$ is the set of constraints explicitly specified in $u$.

- $C_{\imp}$ is a set of implicit constraints the user intends

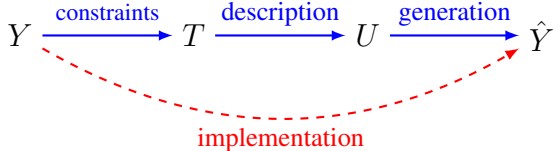

*Figure 2.* Two specification paths. **Natural language path** (solid blue): $Y \to T \to U \to \hat{Y}$. The user forms a task $T$ (allowing indifferent compression), describes it in natural language $U$ (allowing pragmatic compression), and the model generates $\hat{Y}$. **Formal language path** (dashed red): $Y \to \hat{Y}$ via direct specification.

(i.e., expects in this context) but not specified in $u$. Not satisfying $C_{\imp}$ is considered a failure.

A constraint is formally an indicator function $c : M \to \{0, 1\}$ for the set of outputs satisfying the constraint.

**Definition 3.2** (Denotation). The *denotation* of a task is defined as:

$$[\![T]\!] := \{y \in M : \forall c \in C_{\imp} \cup C_{\exp}, c(y) = 1\} \quad (1)$$

## 3.2. The Specification Chain

We model specification as information flow from the user's intended output to the generated result. The key insight is that natural language and formal language follow different paths with different information-theoretic properties. For our purposes, regard the actual output (or implementation) format as the formal language.

**Definition 3.3** (Random Variables). Let $Y$ be a user's envisioned output in meaning space $M$, with prior $p_M$, let $T$ be a task defining an equivalence class $[\![T]\!] \subseteq M$ of acceptable outputs, let $U$ be a natural language description, and let $\hat{Y}$ be the generated output.

We emphasize that $Y$ is a single instance, corresponding to an envisioned or actual outcome, as opposed to $[\![T]\!]$, which is a set of acceptable outcomes. We assume $Y \in [\![T]\!]$, i.e., the user's intent is realizable as a task. Cases where the user envisions an output that no task description can pick out (e.g., self-contradictory) fall outside our framework; in practice, such intents surface as failures during iteration.

Generating $\hat{Y}$ can follow two paths (Figure 2):

**1. Natural Language Path:** $Y \to T \to U \to \hat{Y}$. Rather than specifying $Y$ precisely, the user specifies a task $T$ such that any $\hat{Y} \in [\![T]\!]$ is acceptable, thus applying *indifferent underspecification* (4). Additionally, the description $U$ states only the explicit constraints of $T$, relying on a model to generate $\hat{Y}$ while satisfying the implicit constraints, thus applying *pragmatic underspecification* (3). Importantly, $U$ and $\hat{Y}$ are visible while $Y$ and $T$ are latent.

**2. Formal Language Path:** $Y \rightarrow \hat{Y}$. Formal languages implement $Y$ directly, applying full specification (2).

The natural language path forms a Markov chain. By the Data Processing Inequality (Cover, 1999):

$$I(Y; \hat{Y}) \leq I(Y; U) \leq I(Y; T) \leq H(Y) \qquad (2)$$

Providing bounds on how much information about $Y$ can reach $\hat{Y}$ through the natural language path.

### 3.3. Task Specificity

We now define the central quantity that determines when each path is advantageous.

**Definition 3.4** (Task Specificity). The *task specificity* $\sigma_\epsilon(T)$ is defined as the minimal mutual information $I(Y; \hat{Y})$ such that $\hat{Y} \in [\![T]\!]$ with probability $> 1 - \epsilon$.

For simplicity, we will usually write $\sigma(T)$ or $\sigma$. Intuitively, $\sigma$ captures how much the task allows the output to diverge from an example one. When $\sigma = 0$, any output is acceptable, and when $\sigma = H(Y)$, only the exact intended output is acceptable. For uniform distributions, this captures exactly the difference in (log) size between the set of possible outputs and the acceptable ones. For example, creative tasks like "generate a poem" have low specificity, since a poem the user had in mind is only loosely indicative of a random acceptable poem. In contrast, reproducing a specific image has maximal specificity.

### 3.4. The Translation Gap

Natural language is optimized for open-ended communication across all domains (§2). When used to describe outputs from a specific domain $M$, this generality incurs a cost.

The expected bits required to describe an output $Y \sim p_M$ using natural language distributed as $p_{\mathrm{NL}}$ is the cross-entropy:

$$H(p_M, p_{\mathrm{NL}}) = H(p_M) + D_{\mathrm{KL}}(p_M \| p_{\mathrm{NL}}) \qquad (3)$$

**Definition 3.5** (Translation Gap). The *translation gap* is $\Delta_{\mathrm{trans}} = D_{\mathrm{KL}}(p_M \| p_{\mathrm{NL}})$, the KL divergence from the domain distribution to the natural language distribution.

The translation gap quantifies the cost of using a general-purpose language to describe domain-specific content. It is always non-negative and is zero only when natural language perfectly matches the domain distribution. $\Delta_{\mathrm{trans}}$ thus represents the irreducible inefficiency of using language modeling as compression (Deletang et al., 2024).

As an example, consider reporting the color of a banana. Bananas in images are typically yellow, but "yellow banana" is marked (i.e., less common) in language because the modifier is redundant (Paik et al., 2021). The translation gap captures this mismatch between visual and linguistic distributions.

The following lemma shows that supporting diverse domains increases the expected translation gap—the more open-endedness, the larger the gap to a random domain.

**Lemma 3.6.** *Let $\mathcal{D}$ be a set of domain distributions. The expected translation gap satisfies:*

$$\mathbb{E}_{p \in \mathcal{D}}[D_{KL}(p \| p_{NL})] = D_{KL}(\bar{p} \| p_{NL}) + JSD(\mathcal{D}) \qquad (4)$$

*where the expectation is for a uniform distribution over $\mathcal{D}$, $p_{NL}$ is a common language, $\bar{p}$ is the average distribution, and $JSD(\mathcal{D})$ is the Jensen-Shannon divergence.*

*Proof.* See Appendix B. □

### 3.5. Specification Costs

We now characterize the cost of each specification path.

**Definition 3.7** (Description Length). Let $L_{\mathrm{NL}}(\sigma)$ be the minimum expected description length to achieve $I(Y; \hat{Y}) \geq \sigma$ via natural language, and $L_{\mathrm{formal}} := \mathbb{E}_Y[L_{\mathrm{formal}}(Y)]$ be the expected length of a formal specification of $Y$.

**Proposition 3.8** (Natural Language Cost). *To achieve $I(Y; \hat{Y}) \geq \sigma$ via the natural language path:*

$$L_{NL}(\sigma) \geq \sigma + \Delta_{trans} \qquad (5)$$

*Proof.* By (2), $I(Y; U) \geq I(Y; \hat{Y}) \geq \sigma$. The description $U$ must therefore carry at least $\sigma$ bits about $Y$. Encoding this in natural language incurs the translation gap, giving the bound. □

The natural language cost can thus be divided into two components: the specificity requirement $\sigma$ (how much information must be conveyed) and the translation gap $\Delta_{\mathrm{trans}}$ (the overhead of using a general-purpose language).

When using formal language, we define redundancy as a measure of the overhead of a formal language beyond the information-theoretic minimum.

**Definition 3.9** (Redundancy). We define the *redundancy* of a formal language as $R_{\mathrm{formal}} = L_{\mathrm{formal}} - H(Y)$.

Formal languages specify $Y$ completely, regardless of how much of that specification the task requires. The expected length therefore decomposes as $L_{\mathrm{formal}} = H(Y) + R_{\mathrm{formal}}$: the entropy $H(Y)$ is the information-theoretic minimum, and $R_{\mathrm{formal}}$ is the redundancy of the encoding. Crucially, no translation gap is incurred.

### 3.6. The Specificity Crossover

We can now state our main result: natural language and formal language dominate in different specificity regimes.

**Theorem 3.10** (Specificity Crossover). *For $Y \sim p_M$, there exists a crossover threshold:*

$$\sigma^* = H(Y) + R_{formal} - \Delta_{trans} \qquad (6)$$

*such that:*

- *For $\sigma(T) < \sigma^*$: Natural language can be more efficient (i.e., possibly $L_{NL} < L_{formal}$)*

- *For $\sigma(T) \geq \sigma^*$: Natural language cannot be more efficient (i.e., necessarily $L_{formal} \leq L_{NL}$)*

*Proof.* By 3.8, achieving $\sigma$ requires $L_{\text{NL}}(\sigma) \geq \sigma + \Delta_{\text{trans}}$. Formal specification costs $L_{\text{formal}} = H(Y) + R_{\text{formal}}$ regardless of $\sigma$. Setting $L_{\text{NL}} < L_{\text{formal}}$ yields:

$$\sigma + \Delta_{\text{trans}} < H(Y) + R_{\text{formal}} \Rightarrow \sigma < \sigma^* \qquad (7)$$

$\square$

The crossover point $\sigma^*$ marks where underspecification benefits can no longer compensate for the translation gap. Below $\sigma^*$, natural language can win by not specifying what doesn't matter or what can be inferred. Above $\sigma^*$, formal language wins by avoiding translation overhead.

*Remark* 3.11. For the threshold $\sigma^*$ to be meaningful, we need $R_{\text{formal}} < \Delta_{\text{trans}}$. This criterion is satisfied whenever $R_{\text{formal}}$ is small (efficient formal language) and $\Delta_{\text{trans}} > 0$ (domain differs from general language). If $\Delta_{\text{trans}} \leq R_{\text{formal}}$, natural language could be more efficient even at full specification by removing redundancy in the target. However, intuitively, we expect domain-specific languages to be better optimized for their domains compared to natural language.

*Remark* 3.12. Proposition 3.8 describes a lower bound, assuming no redundancy in natural language. In practice, natural language also has redundancy (Shannon, 1951), even if lower than in some formal languages (Hindle et al., 2016), thus tightening the bound.

*Remark* 3.13. Our analysis assumes a one-shot setting, in which a valid generated instance is obtained. In practice, pragmatic inference errors often occur and latent preferences often arise, resulting in an iterative process of trial and error. This process can involve repetition and overspecification raising the price of natural language and lowering the threshold.

*Remark* 3.14. In the other direction, using formal languages typically requires domain learning and mental effort (Casalnuovo et al., 2018). These factors present additional costs that may raise $\sigma^*$, but should not eliminate the crossover.

An important conclusion from the framework and theorem is that if a task is compositional (§2.3), different components can vary in their implementation method. This idea leads to *hybrid* approaches, combining both natural and formal language.

# 4. Case Study: Image Generation and Other Modalities

Text-to-image generation has emerged as one of the most visible applications of generative AI, with significant implications for artists' workflows and employment (Jiang et al., 2023). Anantrasirichai & Bull (2022) provide a broader examination of AI for creativity across modalities, including ethical considerations, while surveys reveal complex attitudes among both professional and non-professional artists (Tang et al., 2024). Understanding the limitations of natural language for image specification is thus both theoretically important and practically consequential.

**The specificity crossover.** Image generation provides a "clean" domain for demonstrating the specificity crossover. Consider white noise images where pixels are independently and uniformly distributed: such images are maximally incompressible, with maximal entropy and zero redundancy ($R_{\text{formal}} = 0$). Since natural language must assign probability mass to all communicative intents, any nonzero translation gap has $\Delta_{\text{trans}} = D_{\text{KL}}(p_M \| p_{\text{NL}}) > 0$. By Theorem 3.10, a crossover threshold $\sigma^* < H(Y)$ must therefore exist.

**Quantitative Illustration.** To illustrate the information-theoretic limits, consider a $250 \times 250$-pixel RGB image. The total number of possible images is $256^{3 \times 250 \times 250} = 2^{1,500,000}$. Assuming a vocabulary of 32 characters, a string of length $N$ encodes $2^{5N}$ possible strings, so $N \geq 300,000$ characters would be needed—comparable to a novel. While the space of images people actually want is obviously smaller, this demonstration shows a fundamental challenge: as users push toward greater specificity, prompts must grow toward this limit, becoming unwieldy. Consider an artist who envisions a detailed work of art. Image generators cannot practically implement this artwork simply because it will be too hard to describe in full detail.

## 4.1. Empirical Evidence in Image Generation

**Trends in Iterative Prompting.** The theoretical crossover manifests clearly in empirical studies of user interactions with text-to-image systems. Don-Yehiya et al. (2023) compiled a dataset of iterative interactions between users and Midjourney, analyzing how prompts evolve across multiple attempts. Their findings reveal two distinct trends that directly support our framework:

1. **Trend 1: Increasing Verbosity.** Prompts become more verbose—the length and syntactic complexity grow—as interactions become longer. Users "climb down the ladder of specification", representing the conversion of pragmatic underspecification (3)—where users assume the model would infer their intent —

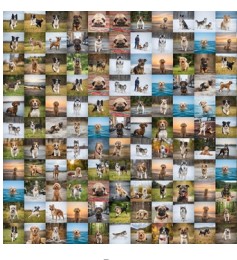 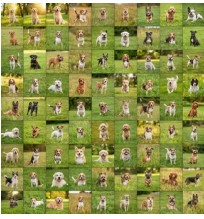 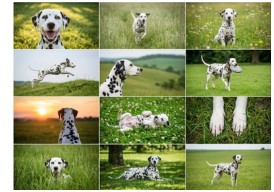 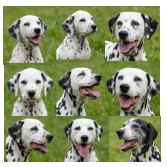 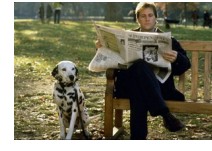

Dogs     Dogs with a grass background     Dalmatians with a grass background

Dalmatians: only a head, visible face, only grass - no sky or flowers or trees

Actor Jeff Daniels as Roger Dearly sitting on a wooden bench in a park .. Pongo the Dalmatian sits patiently on the grass beside him ...

**Low task specificity** ➝ **High task specificity**

*Figure 3.* Illustration of task specificity in image generation. As the description becomes longer, the space of images is narrower. At the limit, if a specific image is desired, perfectly describing it in words is inefficient (compared to filming or drawing). All images were generated by Gemini-3, except for the image from Disney's 1996 *101 Dalmatians* (right).

into full specification (2) or even overspecification (1), where users add redundant information to compensate for misinterpretation.

2. **Trend 2: Increasing Model-Orientation.** Prompts shift toward language patterns that the model responds to more reliably. Perplexity decreases as users adopt "magic words"—terms that empirically improve generation quality regardless of their natural language meaning (e.g., "4k," "artstation," "trending"). This adaptation represents users implicitly reducing the translation gap (§3.4) by shifting toward language that better matches the model's training distribution, sacrificing naturalness for domain-specific efficiency.

**Prompt engineering.** Research on prompt engineering for image generation (Oppenlaender, 2024; Liu & Chilton, 2022) further confirms these findings. Initially, prompts are highly underspecified ("a cat"), efficiently describing large equivalence classes. As users seek more specific outputs, prompt engineering tends to create increasingly specific prompts ("a fluffy orange tabby cat, sitting, front view, studio lighting, photorealistic") (Hao et al., 2023). The phenomenon of *negative prompts* (Ban et al., 2024; Park et al., 2025), such as "no extra fingers," further demonstrates the process: users initially operate at pragmatic underspecification, assuming shared knowledge, but are forced toward overspecification upon failure.

**Hybrid Approaches.** Leveraging compositionality (§2.3), hybrid approaches can benefit from both modalities. For images, this suggests combining natural language for high-level concepts ("professional business portrait") with sketches or masks for spatial layout or specific details. Work on hybrid inputs (Zeng et al., 2022; Vinker et al., 2025) demonstrates this principle. As task specificity increases, more specification naturally shifts toward direct manipulation in the target domain, with natural language reserved for aspects where underspecification provides genuine value.

## 4.2. Additional Modalities

**Code Generation.** In code generation, the distinction between specification understanding and code optimization becomes apparent. We generally divide code requirements into two broad categories: outcome-related, which can include outputs and runtime, and code-related, which can include readability, maintainability, code weaknesses, etc. Generally, loose code-related requirements allow a large degree of autonomy for the agent, potentially making outcome-related requirements more efficient. Conversely, code-related requirements are often more general than outcome-related ones, since a single function can correspond to infinitely many outcomes, thus being more efficient.

In simple visualization or proof-of-concept tasks, natural language excels: "make a bar chart of sales data" efficiently conveys intent with few code-related requirements and loose outcome-related ones. These tasks leverage indifferent underspecification (4) to avoid the overhead of precise coding. However, production code requirements dramatically increase task specificity. Outcome-related requirements about edge cases, etc., and code-related requirements for error handling, optimization, and security represent implicit constraints that surface through testing. Additionally, errors in production code are more costly, forcing extra specification. Multiple works show that while AI coding assistants accelerate initial development, they can lead to low-quality code (Li et al., 2024), requiring iterative refinement that often negates these gains (Becker et al., 2025). These trends align with our crossover prediction: as task specificity increases, the cross-entropy cost exceeds underspecification benefits.

The hybrid approach is also prominent in coding (Barke et al., 2023; Tang et al., 2025). Modern coding assistants providing intelligent code completion can achieve compression benefits through pragmatic underspecification (3; exploiting repeated patterns, suggesting contextual completions), while partially avoiding the overhead of natural language. Moreover, when using coding agents iteratively, the input

consists of code and new requirements (or error reports), which can be seen as a hybrid setting (at least when the user gives some form of feedback, even implicit, about the code itself).

**Audio and Music Generation.** Text-to-audio generation (Agostinelli et al., 2023; Lerch et al., 2025) shows similar patterns. For general requests like "relaxing piano music", the large range of acceptable outputs allows efficient natural language description, exploiting vast underspecification. Conversely, requesting specific output can grow extremely verbose, at which point, MIDI specification or musical notation—formal languages optimized for concrete musical description— become more efficient (Mitra & Zualkernan, 2025). Here too, hybrid methods that guide generation with musical examples were proposed (Zhu et al., 2024).

**Text Generation and Summarization.** Interestingly, the crossover can favor description in some text generation tasks. Writing prose from bullet points (summary → full text) may be more efficient than direct prose writing because the bullets provide compressed details while underspecifying stylistic choices. Similarly, in story generation (Teleki et al., 2025) and summarization (Zhang et al., 2025), loose requirements (e.g., what exactly should be included in the summary?) that allow a large output space (Chawla et al., 2026) can utilize underspecification. As the requirements become more precise, the ability to create a satisfying outcome based on a prompt alone diminishes. In these cases, hybrid approaches are natural, where partial input (Lin et al., 2025; Lee et al., 2025; Urlana et al., 2024) or post-generation editing of the outcome can combine both qualities.

## 5. Implications and Future Directions

Our analysis reveals that natural and formal languages occupy different niches in the specification space. Natural language excels at low specificity tasks with large equivalence classes, creative tasks where exploration is valued, and contexts with low expertise requirements, where underspecification can be exploited Formal languages excel at high specificity tasks, contexts where domain expertise is available, and production systems requiring precision and verifiability. Formal languages succeed through domain-tailored language, lower error rates, and formal verifiability. We reiterate the classical view (Meyer, 1993; Wing, 2002; Gervasi & Nuseibeh, 2002; Ferré, 2016) that formal specification should be complementary to natural descriptions.

Formal languages are also crucial for a precise understanding of the area of interest, allowing deep insight into underlying rules, which is also beneficial in practice. As Dijkstra observed, formal texts are "an amazingly effective tool for ruling out all sorts of nonsense that, when we use our native tongues, are almost impossible to avoid" (Dijkstra, 1978).

Our framework suggests that current evaluation practices may mischaracterize the value of many tasks. For example, popular code generation benchmarks, such as HumanEval (Chen et al., 2021) and SWE-bench (Jimenez et al., 2023), primarily test code implementation rather than specification understanding. Benchmarks for specification understanding and hybrid programming should also be developed. Specifically, code benchmarks should explicitly compare: natural language specification with AI generation, code completion assistance within formal languages, and naive coding without assistance, across varying task specificity levels, including both outcome- and code-related requirements.

Regarding practical tools, we advocate for hybrid systems, optimized for pragmatic understanding, that allow seamless user involvement in the process (Tang et al., 2025). These systems will benefit from both qualities: efficient underspecification and pragmatics of natural language, and precise specification of formal language. Our analysis also frames two open empirical questions: how should systems balance pragmatic assumptions against asking for clarification, and at what compositional boundaries should formal components be optimally used? Both can be studied directly within the specificity-crossover framework. Importantly, in these systems, human expertise will still have a significant role, allowing better integration and supervision.

## 6. Alternative Views

The position we argue against holds that advances in LLMs will eventually render formal languages obsolete. This position is clearly present in the industry, as mentioned in the Introduction (§1). For example, in code, proponents point to rapid improvements in code generation and the democratizing potential of natural language interfaces (Jiang et al., 2024), viewing current limitations as engineering challenges that scale and better training will overcome. Our analysis challenges this view not by denying continued progress, but by identifying fundamental information-theoretic limits: the translation gap $\Delta_{\text{trans}}$ represents structural constraints that persist regardless of model capability. Even a perfect model cannot escape the fact that natural language, optimized for open-ended communication, incurs unavoidable overhead when specifying concrete details in restricted domains.

We also argue against the opposite view, saying that tools such as code generators are "more harm than good". This sentiment is non-negligible among programmers (as evinced in the survey mentioned in §1). We argue that despite some challenges, these tools have a theoretical role through indifferent and pragmatic underspecification, providing value for both professional and low-level programming.

# 7. Conclusion

We presented a formal framework that shows natural and formal languages serve fundamentally different purposes in task specification. Our framework builds on linguistic insights, specifically underspecification. Under this framework, natural language achieves efficiency through underspecification—both indifferent and pragmatic—while formal languages provide precise control. We show that, due to the translation gap, a specificity crossover point exists between cases where pure natural language or pure formal language is more efficient. Hybrid models can achieve both benefits by applying different methods across components. We also demonstrated that our framework is applicable to other modalities, such as images, code, audio, and even text.

Our central claim is the following: natural language will not—and should not—fully replace formal languages. Rather, both languages form a complementary toolkit, with optimal choice depending on task specificity, novelty, and professional requirements. Instead of abandoning formal languages, we should invest in tools that improve efficiency within formal domains while reserving natural language for areas or components where underspecification provides genuine value. The future lies in seamless integration of both modalities, allowing users to operate fluidly across the specification spectrum.

We conclude with Jensen Huang's broader claim:[5]

> AI is the great equalizer... For the last 50 years, 60 years, Computer Science became a field of science, and it was available to tens of millions of people out of billions of people. This technology was hard to use. ... but now, all of a sudden, there's a new programming language. This new programming language is called "human". ... The way you program a computer today [is] to ask the computer to do something for you, even write a program, generate images, write a poem. Just ask it nicely.

Code and image generators provide an extremely powerful tool, especially for non-experts. However, we argue that the role of domain experts—such as programmers and artists—will not become obsolete.

# Acknowledgments

This research was supported by grants from the Israeli Ministry of Science and Technology, the Council for Higher Education, and the Israel Science Foundation (Grant No. 2424/21).

---

[5]London Tech, June 2025; https://youtu.be/SsNS3Xm9ig4?t=1940

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

## A. Imperatives and Autonomy

Commands and requests highlight the role of listener autonomy in underspecification. Sperber & Wilson (1986) argue that communication is not passive decoding but involves active inference about relevance to the listener's goals, emphasizing the listener as an agent. Barker (2010) models choice in imperatives (e.g., "take an apple or an orange") with logic that accounts for the listener's resources and preferences.

Autonomy in communication can be understood through the ladder of specification: *Strong autonomy* corresponds to indifferent underspecification: the speaker genuinely does not care which option the listener chooses and grants full discretion. *Weak autonomy* corresponds to pragmatic underspecification with delegation: the speaker has implicit preferences or trusts the listener to act in the speaker's interest, but does not explicitly constrain the choice. This distinction has practical implications for human-AI interaction: systems should distinguish between cases where user indifference grants genuine autonomy versus cases where the system should infer and follow implicit user preferences.

## B. Proof for Lemma 3.6

*Proof.* Let $\mathcal{D} = \{p_1, p_2, \ldots, p_n\}$ be a set of $n$ domain distributions, and let $\bar{p} = \frac{1}{n} \sum_{i=1}^{n} p_i$ denote the centroid (average) distribution. We assume a uniform distribution over $\mathcal{D}$.

The Jensen-Shannon divergence of the domain set is defined as:

$$\text{JSD}(\mathcal{D}) = H(\bar{p}) - \frac{1}{n} \sum_{i=1}^{n} H(p_i) \tag{8}$$

where $H(\cdot)$ denotes the Shannon entropy.

We begin by expanding the expected KL divergence:

$$\mathbb{E}_{p \in \mathcal{D}}[D_{\text{KL}}(p \| p_{\text{NL}})] = \frac{1}{n} \sum_{i=1}^{n} D_{\text{KL}}(p_i \| p_{\text{NL}})$$

$$= \frac{1}{n} \sum_{i=1}^{n} \left( -H(p_i) - \sum_x p_i(x) \log p_{\text{NL}}(x) \right) \tag{9}$$

Rearranging terms:

$$= -\frac{1}{n} \sum_{i=1}^{n} H(p_i) - \frac{1}{n} \sum_{i=1}^{n} \sum_x p_i(x) \log p_{\text{NL}}(x)$$

$$= -\frac{1}{n} \sum_{i=1}^{n} H(p_i) - \sum_x \left( \frac{1}{n} \sum_{i=1}^{n} p_i(x) \right) \log p_{\text{NL}}(x)$$

$$= -\frac{1}{n} \sum_{i=1}^{n} H(p_i) - \sum_x \bar{p}(x) \log p_{\text{NL}}(x) \tag{10}$$

Now, we add and subtract $H(\bar{p})$:

$$= H(\bar{p}) - \frac{1}{n} \sum_{i=1}^{n} H(p_i) - H(\bar{p}) - \sum_x \bar{p}(x) \log p_{\text{NL}}(x)$$

$$= \text{JSD}(\mathcal{D}) + \left( -H(\bar{p}) - \sum_x \bar{p}(x) \log p_{\text{NL}}(x) \right)$$

$$= \text{JSD}(\mathcal{D}) + D_{\text{KL}}(\bar{p} \| p_{\text{NL}}) \tag{11}$$

which completes the proof. □

