# OpenReview forum: "Position: Natural Language Should Not Fully Replace Formal Languages"
_ICML.cc/2026/Position_Paper_Track — ICML 2026 Position Paper Track regular_

### Official Review · Reviewer_fdKi · 2026-02-19

**Significance:** 2
**Argument Clarity:** 3
**Rating:** 4
**Confidence:** 4

**Questions:**

1. Can you elaborate on the difference between Y and T and [T]?

I roughly sensed that they are two things but did not fully understand.

2. Can you elaborate on the difference between $L_{formal}$ and $H(Y)$?

The term $L_{formal}$ first showed up in Definition 3.9 without conceptual foreshadowing. It seems to me that there is a circular definition between $R_{impl}$ and $L_{formal}$. It would be better if the difference between $L_{formal}$ and $H(Y)$ is first described, then summarized in the Eq(6), rather than stating the equation first and imply that $L_{formal}$ and $H(Y)$ are different because of the equation.

Based on my pragmatic inference, I believe that $L_{formal}$ and $H(Y)$ differ because $L_{formal}$ does not leverage the fact that you can assign shorter descriptions to more frequent instances and longer descriptions to less frequent ones. For example, an image is specified as an $H \times W \times C$ tensor regardless. But optimal compression may leverage something like Huffman coding to use less bits for images with higher likelihood. So my suggestion is to make such arguments explicit before introducing Definition 3.9 and Proposition 3.10.

**Alternative Views Section:**

Yes

**Compliance With Llm Reviewing Policy A Conservative:**

Affirmed.

**Discussion Potential:**

3

**Final Justification:**

The rebuttal addressed my concerns about vague call-to-actions, and lack of elaboration in alternative views. It also addressed my questions about the exact meaning of notations. So I improved my rating to 4.

**Paper Summary:**

This paper argues that natural language (NL) and formal language (FL) play distinct and complementary roles. Thus, **one cannot replace the other**, despite recent opinions in the LLM/code-generation community that favors a domination of NL.

The arguement is established via the lens of an **information-theoretic tradeoff** — NL excels at underspecification. FL is good at specification. Forcing the content which originally favors FL to be translated into NL is in fact information-theoretically inefficient.

The paper calls to **adopt a hybrid view**, rather than leaning towards either "NL replaces FL" or "FL replaces NL". This hybrid view can be applied **in many modalities**:
- image: combining NL high-level concepts with sketches or masks for spatial layout.
- code: code completion can be retreated as prompting with code (FL) complementary to prompting with NL for what the code does.
- music: MIDI specification counts as FL.
- text: partial text input or a prose outline counts as FL

**Position:**

Yes

**Position In Title:**

Yes

**Related Work:**

4

**Strengths And Weaknesses:**

### Strengths

1. The formal framework introduced in $\S3$, grounded in information theory and linguistics, serves as **a novel scaffold to think about the prompt → generation pipeline as a multi-stage translation**. This is a neat framework and may seed a rich set of future analytical, empirical, and development works.
2. The two paths for goal specification (NL path, FL path) is very illuminating. They show two sources of inefficiency: one is **translation overhead** in the NL path, the other is **redundancy** in the FL path. The tension between these two types of inefficiency induce the **crossover** and the tradeoff. This is a novel lens to substantiate the position.


### Weaknesses

**General comment on weaknesses**: my current rating is rejection not because I do not see a merit of this paper, but because I see lots of potential merits which haven't yet been adequately articulated or substantiated. Thus I write down my suggestions here to help improve this paper.

**1. The call-to-actions are less elaborated.**

The advocacy for a hybrid approach is a valid over-arching proposal. I believe the paper can be greatly improved if subcomponents of this proposal is further articulated. I can directly see many fruitful extensions. In particular, you might want to leverage the specification ladder, or the $Y \rightarrow T \rightarrow U \rightarrow \hat{Y}$ pathway, as a conceptual anchor.

For example, you can advocate *detection* of the crossover over the course of communication —— as the user input becomes increasingly verbose, a system may detect it and actively ask “do you want to send me a sketch?"

For another example you can advocate *estimation* of the crossover ahead of time —— if a system allocates small effort in advance to figure out whether the upcoming task is a low-specificity or high-specificity one, then it might be able to actively steer the mode of human-computer interaction by “sliding” along the specificity spectrum.

The two paths formalized in $\S3$ are very interesting. However, most of the equations correspond to a hypothesis rather than an existing result of theoretical investigation. For example, Eq2 is intuitively true, but we don’t clearly know how much information is lost at each stage along the way ($Y \rightarrow T \rightarrow U \rightarrow \hat{Y}$). For another example, we also don’t know how $R_{impl}$ can be operationalized and measured. You could imagine the task fixed as song generation, but there is a user spectrum from lay-people to musicians. Then there could be a curve tracking the variation of $R_{impl}$ along the expertise spectrum. What would such a curve look like? Visualizing such a curve will offer profound insights, but it requires us to invent a way to measure or estimate $R_{impl}$.

In summary, I raised this weakness but I suggest reframe it as a strength (which needs additional articulation) because the formalization ($\S3$) is a strong contribution, seeding a rich set of future analytical, empirical, and developmental studies.

**2. The alternative-views section is less elaborated.**

The paper currently treats the two extremes --- pure NL and pure FL --- as potential objections. But this seems rather apparent. I come up with two potential alternative views.

(a) Most users aren’t professional, so the use cases will heavily skew towards underspecification where NL wins.

This is indeed true: in terms of the sheer size of the user group, lay users dominate while professionals are minority. So developers are motivated to care about lay-user needs and overlook what professionals may prefer. But in support of this paper’s position, you need to argue for the professionals. You might want to say that the purpose of studying generative systems is not merely building products for the general public. The purpose also lies in advancing dedicated domains —— language models advance linguistics, biosequence modeling advances biology, music generation advances music, image generation advances art … FL and the hybrid approach can be proved most useful in these AI-for-X paradigms.

(b) The argument that NL loses efficiency to FL is only valid when efficiency is measured in terms of description length (i.e. number of bits). However, this largely disregards other types of efforts —— the user’s mental effort, the developer’s engineering effort, the feasibility of integrating domain-specific languages into an established system (e.g. The transformer backbone requires that everything has to be “tokens” and all embedding matrices have to be pretrained).

I personally do not hold this opinion. But I could see it as a potential adversarial view to be addressed.

**Support:**

4

---

> ### Author Rebuttal · Authors · 2026-03-27
>
> We thank the reviewer for the detailed and constructive critique.
>
> **Re. Weakness 1 — call-to-actions and elaboration of the hybrid approach:**\
> Thanks for the suggestions. These two examples—detecting the crossover from increasing prompt verbosity, and estimating it ahead of time to steer the interaction mode—are excellent illustrations of how the framework can be operationalized. These ideas can be partially implemented by simple prompting, but empirical investigation of real data can add substantive value. Our position calls for appropriate benchmarks for evaluation (l403), and we will clarify that this type of data is also valuable for practical improvement. \
> We will present these ideas as directions opened up by the framework rather than contributions of the current paper, since the focus of this paper is the fundamental framework.
>
> **Re. Weakness 1 — operationalizing R_impl and measuring information loss:**\
> We agree that the equations in Section 3 currently serve as theoretical scaffolding. While defining an exact effect of expertise is complicated, we can assume that expertise reduces the cost per unit of formal specification, shifting the threshold. This effect is related to Remark 3.13, and we will make it more explicit. The reviewer's suggestion of fixing a task domain (e.g., song generation) and tracing R_impl across a user expertise spectrum is an intriguing direction for future empirical work. We will frame Section 3 more explicitly as providing the conceptual vocabulary for such future studies.
>
> **Re. alternative view (a) — most users are lay users, so NL wins by volume, but FL has other purposes:**\
> To our understanding, the reviewer here strengthens our position, arguing that the FL has value not only in terms of more production but also in terms of advancing the specific domains of the formal language. This is a valid point, and we will mention it.
> We do emphasize, however, that our position is that formal language is beneficial also in production terms. Additionally, we mention the trend of model-oriented language (Section 4.1), showing that the boundary between NL and FL becomes softer. So even amateurs can, and should, adopt formal elements. Moreover, computer systems have also started to converse with one another in some hybrid of natural and formal language (e.g., agents communicating with other agents), thus, even in terms of volume, pure NL may not be completely dominant.
>
> **Re. alternative view (b) — description length ignores mental effort:**\
> This is a valid argument. We somewhat address this in Remark 3.13 (that formal language requires domain learning), but we will elaborate on it. Importantly, the mental effort should be reduced based on experience and expertise, thus emphasizing the value of experienced programmers.
>
> **Re. alternative view (b) — description length ignores engineering costs:**\
> This case falls under what we describe as pragmatic underspecification (Definition 2.1): the user can omit many details that can be inferred from the context. We mentioned this is a case where NL can benefit the user. However, as requirements become unique, inferring them from context can be difficult (or error-prone), in which case formal language may be necessary.
>
> **Re. Q1 — the distinction between Y, T, and [[T]]:**\
> Y (Definition 3.3) is a single "envisioned" output — for example, an artist who envisions an exact image. In practice, a user in the NL path probably does not always think of a fully detailed example, but any implementation (as in the alternative path) necessarily chooses a single instance. T (Definition 3.1) is the task, defined by constraints and a description. [[T]] (Definition 3.2) is the realization of T—the set of elements in the output space that meet the constraints. Importantly, Y is a specific instance and [[T]] is a set of instances (which includes Y, but may include many more, depending on the specificity). We will clarify these.
>
> **Re. Q2 — the relationship between L_formal and H(Y):**\
> Your understanding is generally correct: H(Y) represents the description length of Y under optimal compression. However, while optimal compression is formal, it is usually uninterpretable even for experts. L_formal represents the description length of a formal language that we consider using (such as code). For example, while bitmaps are tedious to read, their interpretation is straightforward; whereas a zip file is practically impossible to make sense of. Similarly, code can clearly be compressed [1], but compressed code may be very hard to write.\
> Regarding the presentation, we agree and will fix this in the final version.
>
> We hope these responses address the reviewer’s concerns. We are happy to discuss further during the author-reviewer dialogue period.
>
> [1] Hindle, A., Barr, E. T., Gabel, M., Su, Z., & Devanbu, P. (2016). On the naturalness of software. Communications of the ACM, 59(5), 122-131.

---

> > ### Author Rebuttal · Reviewer_fdKi · 2026-04-01
> >
> > Re alternative view (a), (b), and Re Q1, R2, are informative.
> >
> > Re. Weakness 1: I agree that call-to-actions can be expanded along (at least) the following dimensions --- (i) calls for appropriate benchmarks (ii) calls for attempt to operationalize this framework (iii) providing the conceptual vocabulary. I think in the revision, it'd be great to allocation more room to call-to-actions section.
> >
> > I decide to increase my rating to 4.

---

### Official Review · Reviewer_LFGx · 2026-03-12

**Significance:** 3
**Argument Clarity:** 3
**Rating:** 5
**Confidence:** 2

**Questions:**

- Line 057, left column. lingistic -> linguistic.
- Line 392, left column, missing period.
- Remark 3.13. Why not quantify redundancy for natural language costs in the formulation. Is it really negligible?
- Can you quantify the total cost of hybrid methods? Since there is a crossover point between pure natural language or pure formal
language, there should be an approximately optimal point like Pareto optimality? Or, can you at least illustrate this with a toy experiment, which should provide more convincing evidence for future readers? More discussions on this point are appreciated.

**Alternative Views Section:**

Yes

**Compliance With Llm Reviewing Policy A Conservative:**

Affirmed.

**Discussion Potential:**

3

**Final Justification:**

After reading the paper, all the reviews, and the rebuttal carefully, I think this paper presents an important and significant position that elicits attention from the AI community. The trend that more and more people adopt code generation strengthens the value of the paper. However, there are a few weaknesses mentioned by me and other reviewers which I believe are not major and could be easily addressed by the authors in the final version. Consequently, I vote for acceptance but with not very strong confidence.

**Paper Summary:**

This paper presents the position that natural languages cannot fully replace formal languages and they should cooperate together for higher efficiency in the future. The authors put forward a ladder of specificity and crossover threshold to show information-theoretic bounds that compare natural language costs with formal language costs. The authors also discuss future directions and alternative views.

**Position:**

Yes

**Position In Title:**

Yes

**Related Work:**

3

**Strengths And Weaknesses:**

**Strengths**
- The topic is of relevance and importance to the AI and ICML community and seamlessly following the current trend of best practices of LLMs.
- The paper is well written and easy to follow.
- Information theoretic bounds are sound and convincing.

**Weaknesses**
- There are a few typos in the manuscript.
- The empirical results supporting the position are not really sufficient.
- Positions are not highlighted in abstract and introduction. The authors might want to highlight one or two sentences.

**Support:**

3

---

> ### Author Rebuttal · Authors · 2026-03-27
>
> We thank the reviewer for the positive feedback and corrections. We will incorporate them in the final manuscript.
>
> **Re. Remark 3.13 — why not quantify redundancy for natural language?**\
> We agree, of course, that natural language presents redundancy [1], and including it would add a term R_NL to Proposition 3.8. We omitted it because our goal is a lower bound on NL cost, and for so we used a conservative assumption. We will add a note clarifying this in the revised Remark 3.13.\
> We also note that, in the case of programming languages, redundancy estimates are significantly larger than for natural language [2], so the effect of NL redundancy is smaller than that of formal language.
>
> **Re. the cost of hybrid methods:**\
> Our analysis is fundamental, showing that a crossover exists. Our framework has practical consequences, even without discerning an exact crossover spot. For example, we encourage training/prompting models to identify cases in which a natural language description seems insufficient, asking the user for examples.\
> We agree with the reviewer that empirical analysis of this phenomenon is of value. In fact, our position calls for researchers to collect and analyze data from code generation, focusing on ambiguous instructions. We call for appropriate benchmarks for evaluation (l403), but this type of data is also important for a better understanding of the interaction between natural and formal languages. We will clarify this in the final version.
>
> [1] Shannon, C. E. (1951). Prediction and entropy of printed English. Bell system technical journal, 30(1), 50-64.
> [2] Hindle, A., Barr, E. T., Gabel, M., Su, Z., & Devanbu, P. (2016). On the naturalness of software. Communications of the ACM, 59(5), 122-131.

---

> > ### Author Rebuttal · Reviewer_LFGx · 2026-04-01
> >
> > I have read the author rebuttal which has fully addressed my concerns. I still lean towards acceptance for this manuscript.

---

### Official Review · Reviewer_pyBf · 2026-03-12

**Significance:** 3
**Argument Clarity:** 3
**Rating:** 4
**Confidence:** 4

**Questions:**

1. If AI systems can resolve ambiguity interactively (asking clarifying questions rather than guessing), how does this affect $\sigma^*$? The crossover analysis assumes single-shot specification, but interactive clarification could fundamentally change the cost comparison.

2. Prompt engineering patterns like "4k, artstation, trending" function as a domain-specific vocabulary emerging within natural language. Does this represent the NL/formal boundary shifting over time, rather than a fixed structural limit as the paper suggests?

3. The paper's motivating case is code generation, but the empirical evidence is strongest for image generation. Is there concrete user data (analogous to the Midjourney analysis) showing the crossover phenomenon in coding workflows?

**Alternative Views Section:**

Yes

**Compliance With Llm Reviewing Policy A Conservative:**

Affirmed.

**Discussion Potential:**

3

**Final Justification:**

The authors have fully addressed my concerns.Therefore, I maintain my positive score.

**Paper Summary:**

This paper argues that natural language cannot fully replace formal languages, even as LLMs advance. The authors formalize this through task specificity and prove a specificity crossover theorem: below a threshold $\sigma^*$, natural language is more efficient via underspecification; above it, formal languages necessarily win. A ladder of specification (from indifferent underspecification to overspecification) provides the conceptual backbone. Case studies across image generation, code, audio, and text support the argument, leading to a call for hybrid systems rather than full natural language replacement.

**Position:**

Yes

**Position In Title:**

Yes

**Related Work:**

3

**Strengths And Weaknesses:**

**Strengths**

1. The paper provides a principled counterargument to the "NL replaces everything" narrative, rooted in a fundamental property of natural language: underspecification. The very reason NL is powerful (you can omit what doesn't matter) is exactly why it is weak (you also omit what does matter).

2. The ladder of specification offers a practical design principle: where the user sits on this spectrum should determine the interface, pointing toward hybrid systems that let users move between NL and formal language depending on specificity.

3. The paper gives a compelling explanation of why prompt engineering exists: users descending the ladder as they hit natural language's limits. This reframes prompt engineering not as a quirky practice but as a natural consequence of information-theoretic constraints.

4. The observation that current code benchmarks (HumanEval, SWE-bench) test implementation but not specification understanding identifies a genuine blind spot.

**Weaknesses**

1. The framework models specification as a single-shot process (user writes description, model generates output), but real specification is iterative and interactive. Users write NL, see output, refine, switch to code for some parts, and go back. More importantly, if AI systems can ask clarifying questions to resolve ambiguity, the effective cost of NL specification at high specificity drops substantially. The single-shot framing may systematically overestimate NL's disadvantage above $\sigma^*$.

2. Code generation is the paper's motivating example (Jensen Huang quote, Stack Overflow data), yet it has the weakest empirical support among the case studies. The image generation analysis is grounded in concrete Midjourney user data, but the code generation discussion relies mostly on general observations about production code quality. Given that code is the most consequential domain for this argument, stronger evidence here would significantly strengthen the paper.

3. The clean binary between NL path and formal path does not reflect how users actually work with LLMs. A prompt like "write a Python function that sorts using merge sort with O(n log n)  guarantee" already mixes NL and formal concepts seamlessly. Similarly, prompt engineering patterns ("4k, artstation") are arguably a domain-specific language emerging within NL. The boundary between natural and formal languages is blurring in practice, and the framework does not account for this.

**Support:**

3

---

> ### Author Rebuttal · Authors · 2026-03-27
>
> We thank the reviewer for the thorough and constructive feedback.
>
> **Re. Weakness 1 and Q1 — iterative and interactive specification:**\
> For simplicity, we analyze a single generation step. We can consider two cases of iterative specification. In one, a final prompt is reached after multiple iterations, a case that still fits into the single-step framework. We do mention the possibility of errors (Remark 3.12), which may require more specific instructions and more iterations, thus lowering the crossover threshold. However, for simplicity, we did not address it in detail. We will add a discussion on this effect.\
> Another case is where the output from previous steps becomes part of the prompt for the next step (e.g., “add X to the code”, “remove X from the background of the image"). This case falls under hybrid language, which we mention multiple times (e.g., in the end of Section 3).\
> We will clarify these points.
>
> **Re. Weakness 2 and Q3 — weaker empirical support for code:**\
> We motivate our work through code generation since it is rapidly evolving and sparking active debate. However, for the same reason, up-to-date data, which represents the most recent code generation methods, is not publicly available yet. Models are constantly being updated and improved. Additionally, modern code generation interactions tend to be highly complex, with multiple turns and a long context that is often unshareable (the codebase). \
> The image generation domain is relatively more stable and is easier to formally analyze due to the simplicity of its basic formalism: matrices of numbers. We use this domain to illustrate the fundamental properties more intuitively and reason about other formal languages.\
> We do note that our position calls for exactly this—we encourage researchers to collect and analyze data from code generation, focusing on ambiguous instructions. We call for appropriate benchmarks for evaluation (l403), but, as the reviewer mentions, this data is also important for a better understanding of the interaction between natural and formal languages.
>
> **Re. Weakness 3 and Q2 — blurring of the NL/formal boundary:**\
> We agree that the boundary is not sharp in practice, and this aligns with our arguments about hybrid languages. The prompt "write a Python function that sorts using merge sort with O(n log n) guarantee" is a hybrid utterance, using NL framing for high-level intent and formal concepts for precise constraints. Similarly, a "4k, artstation" prompt patterns combine natural language with formal elements, reducing the translation gap through domain adaptation, consistent with the model-orientation trend we mention in Section 4.1. The blurring of the boundary reflects users and systems co-adapting toward efficient hybrid encodings, which we argue can benefit both from underspecification when relevant and from full- or over- specification when necessary.

---

> > ### Author Rebuttal · Reviewer_pyBf · 2026-04-03
> >
> > My concerns are adequately addressed. The boundary blurring point (W3) supporting the hybrid position is convincing, and the interactive specification decomposition (W1) is reasonable.

---

### Official Review · Reviewer_XdqB · 2026-03-16

**Significance:** 3
**Argument Clarity:** 3
**Rating:** 5
**Confidence:** 3

**Questions:**

Check the weakness part for questions about "languages".

**Alternative Views Section:**

Yes

**Compliance With Llm Reviewing Policy A Conservative:**

Affirmed.

**Discussion Potential:**

4

**Final Justification:**

Cool

**Paper Summary:**

This paper argues that formal languages are still valuable and complementary to natural languages, even assuming a "perfect" LLM. They analyze the effects of languages in task specification as information-theoretic limitations. They proved and showed practical evidence (e.g., in text-to-image) that there exists a threshold beyond which the cost to express formal requirements into natural language exceeds the cost of direct formal specification. This threshold is related to the task's goal (how specific the descriptions need to be), the task's distribution (how close the task is to natural language domains), and the formal languages' properties (how efficient they are for specifying the task). They further advocate for the combinations of formal and informal languages, such as sketches (formal) and texts (informal) for image generation.

**Position:**

Yes

**Position In Title:**

Yes

**Related Work:**

3

**Strengths And Weaknesses:**

This paper is enjoyable to read. The position is clean, well-supported, and easy to understand. It analyzes the pros and cons of current popular languages regarding task specifications through the information-theoretic perspectives. The practical evidence from the image, audio, and code domains is interesting and persuasive. I think it is very likely to inspire discussions, and it is timely to discuss this topic right now.

This paper could be even better, in my understanding, if it could define and analyze languages in more detail (i.e., in more "formal" languages). What are formal/informal languages? What distinguishes them? How to specify LLMs with lots of prompts, in-context examples, or domain-specific skills? Are there fundamental limitations for languages themselves, or could we build a unified, better language that obtains the benefits of both worlds?

**Support:**

4

---

> ### Author Rebuttal · Authors · 2026-03-27
>
> We thank the reviewer for the positive feedback and comments.
>
> **Re. what counts as formal:**\
> In this paper, we define formal language (l138) based on the ladder of specification—a formal language is a language with no ambiguity at the Question Under Discussion (QUD). This definition does not explicitly define specific symbols or tokens that the language can use or how disambiguation is achieved. We will further elaborate on this.
>
> **Re. prompts, examples, and skills:**\
> Our analysis assumes a “perfect” model—one that does not fail to meet the explicit requirements. Therefore, if the reviewer’s question refers to built-in system instructions and skills, this can be seen as part of the model and will not affect the issue of underspecificity.
> If the question refers to user prompting, then it can be seen as climbing the specificity ladder, towards formal unnatural language or even overspecification.
>
> **Re. a unified language that obtains the benefits of both worlds:**\
> Our framework suggests a fundamental limit to such unification: any language must trade off between open-endedness (requiring underspecification) and domain precision (requiring its resolution).
> However, a hybrid language may achieve both qualities through compositionality, allowing different components of a task to be specified in the most efficient modality. We mention many examples of this in Section 4.

---

> > ### Author Rebuttal · Reviewer_XdqB · 2026-04-04
> >
> > Thanks for the clarification. Built/Trained-in models (such as one with the Imagenet dataset) will perform better on tasks such as drawing a dog (as those in ImageNet). The interpreter/model could also be important as languages and tasks in that sense, if I understand correctly. But I get the high-level idea. It's a cool paper overall.

---

### Decision · Program_Chairs · 2026-04-30

**Decision:**

Accept (regular)

**Comment:**

The paper addresses a very timely point, states a clear position, and provides more elegant and formal arguments than I would have thought possible for its point.  Reviewers commented on how well-written and enjoyable to read it is.  There was detailed discussion between authors and reviewers that persuaded some reviewers to be more positive than originally.